# A Pilot Study of the Predictive Potential of Chemosensitivity and Gene Expression Assays Using Circulating Tumour Cells from Patients with Recurrent Ovarian Cancer

**DOI:** 10.3390/ijms21134813

**Published:** 2020-07-07

**Authors:** Stefano Guadagni, Marco Clementi, Francesco Masedu, Giammaria Fiorentini, Donatella Sarti, Marcello Deraco, Shigeki Kusamura, Ioannis Papasotiriou, Panagiotis Apostolou, Karl Reinhard Aigner, Giuseppe Zavattieri, Antonietta Rossella Farina, Giuseppe Vizzielli, Giovanni Scambia, Andrew Reay Mackay

**Affiliations:** 1Department of Applied Clinical Sciences and Biotechnology, University of L’Aquila, 67100 L’Aquila, Italy; marco.clementi@univaq.it (M.C.); francesco.masedu@univaq.it (F.M.); antonietta.farina@univaq.it (A.R.F.); andrewreay.mackay@univaq.it (A.R.M.); 2Department of Oncology and Hematology, Ospedali Riuniti Marche Nord, 61121 Pesaro, Italy; g.fiorentini2020@gmail.com (G.F.); d.sarti@fastwebnet.it (D.S.); 3Peritoneal Surface Malignancies Unit, Colon and Rectal Surgery, Fondazione IRCCS Istituto Nazionale Tumori, 20133 Milan, Italy; info@marcelloderaco.com (M.D.); shigeki.kusamura@istitutotumori.mi (S.K.); 4Research Genetic Cancer Centre International GmbH, 6300 Zug, Switzerland; papasotiriou.ioannis@rgcc-international.com; 5Research Genetic Cancer Centre S.A, 53100 Florina, Greece; panagiotis.apostolou@rgcc-genlab.com; 6Department of Surgical Oncology, Medias Klinikum GmbH & Co KG, Krankenhausstr. 3a, 84489 Burghausen, Germany; info@prof-aigner.de (K.R.A.); g.zavattieri@medias-klinikum.de (G.Z.); 7Fondazione Policlinico Universitario Agostino Gemelli IRCCS, 00168 Roma, Italy; giuseppevizzielli@yahoo.it (G.V.); giovanni.scambia@policlinicogemelli.it (G.S.)

**Keywords:** liquid biopsies, circulating tumour cells, precision oncotherapy, recurrent ovarian cancer, hypoxic isolated abdominal perfusion

## Abstract

Circulating tumour cells (CTCs) from liquid biopsies are under current investigation in several cancers, including epithelial ovarian cancer (EOC) but face significant drawbacks in terms of non-standardised methodology, low viable cell numbers and accuracy of CTC identification. In this pilot study, we report that chemosensitivity assays using liquid biopsy-derived metastatic EOC CTCs, from 10 patients, nine with stage IIIC and one with stage IV disease, in progression after systemic chemotherapy, submitted for hypoxic isolated abdominal perfusion (HAP), are both feasible and useful in predicting response to therapy. Viable metastatic EOC CTCs (>5 cells/mL for all 10 blood samples), enriched by transient culture and identified by reverse transcription polymerase chain reaction (RT-PCR) and indirect immunofluorescence (IF), were subjected to flow cytometry-based Annexin V-PE assays for chemosensitivity to several chemotherapeutic agents and by RT-PCR for tumour gene expression profiling. Using a cut-off value of >80% cell death, CTC chemosensitivity tests were predictive of patient RECIST 1.1 responses to HAP therapy associated with 100% sensitivity, 50% specificity, 33% positive predictive, 100% negative predictive and 60% accuracy values. We propose that the methodology employed in this study is feasible and has the potential to predict response to therapy, setting the stage for a larger study.

## 1. Introduction

Epithelial ovarian cancer (EOC) is the 7th most common cancer and 8th leading cause of cancer mortality in women, with a cumulative 5-year survival rate below 45% [1], and in cases with regional and distant invasion, 5-year survival rates are approximately 70% and 30%, respectively. The current standard of care for patients with advanced EOC is debulking surgery combined with systemic paclitaxel and carboplatin chemotherapy [2]. However, although patients have a good response to initial treatment, most experience relapse related to factors, including incomplete tumour debulking and presumed acquisition of resistance to platinum-based chemotherapy [3]. EOC patients considered to be platinum-resistant are commonly those who have relapse following a systemic platinum-based chemotherapeutic regimen. This assumption, however, is normally based upon disease-relapse and not molecular evidence, and is used to characterise four subsets based upon platinum free interval (PFI) duration (<1 month, 1–6 months, 6–12 months and >12 months), which correspond to platinum-refractory, platinum-resistant, partially platinum-sensitive and fully platinum-sensitive categories. Since 2015, a new treatment-free interval (TFI) concept has been adopted that also considers the additional factors of 1st line systemic therapy, breast cancer type 1 susceptibility protein (BRCA) mutational status and clinical eligibility for systemic platinum-based chemotherapy. In any case, the standard of care for patients with platinum-sensitive recurrent EOC is to re-treat with systemic platinum-based chemotherapy, whereas patients with platinum-resistant recurrent EOC tumours are treated with systemic non-platinum-based chemotherapy. Poly (ADP-ribose) polymerases (PARPs) are important nucleoproteins involved in DNA damage-repair and PARP inhibitors have demonstrated efficacy, as maintenance therapy, in adult patients with recurrent EOC, who exhibit complete or partial responses to systemic platinum-based chemotherapy [4].

Treatment options for patients with platinum-resistant EOC include single agent chemotherapy with best supportive care, or a range of aggressive, multi-agent therapeutic regimens for asymptomatic patients comprised of taxanes, anthracyclines, gemcitabine, topotecan and/or trabectedin in various combinations and sequences, and non-platinum monotherapy combined with bevacizumab, followed by maintenance therapy, has also recently been approved, following a landmark trial [5]. However, since there is no evidence supporting more than one line of chemotherapy in platinum-resistant relapsed EOC, there is growing consent among patients to undergo novel therapies for modest gains.

Within this context, novel loco-regional chemotherapeutic procedures are under investigation to prolong survival and preserve quality of life (QoL). In stage III EOC, intravenous and intraperitoneal chemotherapeutic combinations prolong overall survival following primary cytoreductive surgery [6,7,8]. However, this approach has been frustrated by problems related to catheterisation and by gastrointestinal and renal side effects, which can be overcome by Hyperthermic Intraperitoneal Chemotherapy (HIPEC) at completion of surgery to maintain therapeutic advantage. Indeed, two recent randomised HIPEC studies reported longer recurrence-free and overall survival times compared to cytoreductive surgery alone [9,10], with the larger study reporting a median recurrence free interval of approximately 14 months in patients with advanced stage stable EOC following systemic carboplatin and paclitaxel chemotherapy, submitted for cytoreductive surgery plus cisplatin HIPEC, followed by systemic carboplatin and paclitaxel chemotherapy [10].

Pressurised intraperitoneal aerosol chemotherapy (PIPAC) is another novel procedure for delivering pressurised aerosols of normothermic chemotherapeutic agents directly into the abdominal cavity. Cisplatin and doxorubicin PIPAC results in objective clinical responses from 62% to 88%, RECIST responses of approximately 50% and median overall survival from 1st PIPAC of approximately 12 months in patients with unresectable recurrent EOC in progression after >2 lines of systemic chemotherapy [11,12].

However, considering that drugs penetrate tumour tissues with HIPEC and PIPAC to depths of 1–3 mm [13] and EOC metastases may be several cm in diameter, Hypoxic Isolated Abdominal Perfusion (HAP) intra-arterial loco-regional drug delivery has been proposed to potentially increase drug penetration. Indeed, cisplatin plus doxorubicin and mitomycin HAP for recurrent stage IIIC and IV EOC, followed by immediate chemo-filtration, has been reported to result in median survival times of 10 and 12 months, respectively, and an improved QoL [14].

In addition, precision oncotherapy based upon tumour chemosensitivity assays has been under evaluation as an alternative therapeutic approach for the treatment of platinum-resistant EOC. Empiric therapies are chosen from the current literature based upon outcomes achieved for a particular tumour-type with single and combinations of chemotherapeutic agents, whereas drug-selection based upon chemosensitivity assays takes cues from the sensitivity of tumour tissues, tumour cell cultures [15,16,17,18,19,20,21,22,23,24,25,26,27] or purified circulating tumour cells (CTCs) to a panel of chemotherapeutic agents in in vitro cytotoxicity assays. In addition to these approaches, important information predicting a potential drug-response can also be gleaned from standard immunohistochemical, gene expression and transcription profiling of non-viable tumour tissues.

With respect to CTCs, flow cytometry studies have detected CTCs with a high degree of sensitivity and specificity in blood samples from melanoma, breast, prostate, pancreatic and colon carcinoma patients [28]. As CTCs represent potential metastatic precursors, CTC purification provides a unique opportunity to gain a more accurate assessment of the genomic, transcriptomic and chemosensitivity characteristics of individual tumours and has prompted the development of several purification techniques based on tumour cell density, size, deformability and biological properties that facilitate positive or negative label-dependent immunoaffinity purification, which now include an antibody-based herringbone-chip CTC purification technique [29,30], with EOC CTCs identified by immunocytochemistry (ICC), reverse transcription polymerase chain reaction (RT-PCR) and fluorescent in situ hybridisation (FISH) for novel gene fusions [31,32]. In spite of no standardised methodology for CTC purification, characterisation or enrichment, CTC quantification and gene/protein expression profiling are already used to predict disease progression and select treatment strategies [33]. However, PCR analysis precludes additional analysis of living CTC and antibodies for CTC purification against epithelial cell-specific proteins such as epithelial cell adhesion molecule (EpCAM) would not purify metastatic CTC populations that have undergone epithelial to mesenchymal transition, which no longer express typical epithelial cell markers [34].

Despite these drawbacks, molecular EOC CTC profiling has unveiled a myriad of biomarkers of potential diagnostic relevance [30]. A study of EOC CTC heterogeneity [35] has identified EpCAM (epithelial cell adhesion molecule), WT1 (Wilms’ tumour protein), MUC16 (cancer antigen 125), MUC1 (cell surface associated protein), KRT7 (cytokeratin-7), KRT18 (cytokeratin-18) and KRT19 (cytokeratin-19) as highly specific markers that associate with tumorigenicity. Kuhlmann et al. [36] in a study more applicable to clinical care, confirmed that CTC expression of the DNA-platinum adduct repair gene ERCC1 (excision repair cross-complementation group 1), was an independent predictor of platinum resistance (OR = 8.5; 95% CI, 1.7–43.6; *p* = 0.010).

Considering that not all treatments are available in every institution and no single treatment strategy fits all, for reasons of lesion size and number, anatomical location, regional lymph node involvement, distant metastases, biomolecular aspects, concomitant disease and previous therapy, it is our considered opinion that treatment strategies for advanced EOC should be multidisciplinary and could benefit greatly from detailed biomolecular characterisation and chemosensitivity assessment of liquid biopsy derived CTCs from individual patients, already USA Food and Drug Administration (FDA)-approved for prognostics [37]. Furthermore, despite the current lack of methodological consensus, CTC-based analytical methodologies are already under investigation for selecting therapeutic strategies in different cancers, including EOC [30,35].

Recently, we reported a method for liquid biopsy predictive oncotherapy based upon purified metastatic CTCs, transiently cultured in vitro, permitting both gene expression profile and chemosensitivity analysis [38,39,40]. Here, we report a pilot study of metastatic CTCs purified from a homogeneous group of stage IIIC and IV EOC patients, presenting with disease relapse in progression after surgery and two lines of systemic chemotherapy, submitted for locoregional HAP chemotherapy, with the aim of confirming the feasibility and utility of assessing chemosensitivity and biological characterisation of liquid biopsies-derived purified CTCs, as a predictive test for selecting appropriate drugs for HAP and for further therapeutic strategies.

## 2. Results

### 2.1. Biological and Clinical Characteristics of 10 Advanced EOC Patients

Table 1 reports biological and clinical characteristics, RECIST 1.1 tumour responses, and survival of 10 advanced stage EOC patients submitted for multidisciplinary treatments.

### 2.2. Chemosensitivity and Tumour Gene Expression Assays Using EOC CTCs

Viable metastatic EOC CTCs (>5 cells/mL; median 9.4 cells/mL and interquartile range 8.2–9.6) (Figure 1), were isolated from liquid biopsies from 10 patients with recurrent EOC and CTC chemosensitivities presented in Table 2. With respect to the two drugs used for HAP, a chemosensitivity cell-death cut-off value of >80% was chosen. This value was achieved in EOC CTCs from six patients who subsequently received both chemotherapeutic agents. Three patients with CTC chemosensitivity values of >80% for one but not the other drug, received the agent achieving cut-off and received a second agent that did not achieve the cut-off, based on a multidisciplinary decision. The remaining patient, for whom chemosensitivity assay cut-off values failed to reach >80%, received the drug pair inducing the highest levels of cell-death. Based upon responses to previous systemic platinum-based chemotherapy, one patient was platinum-resistant, six were partially platinum-sensitive and three were fully platinum-sensitive. Chemosensitivity assays confirmed ≥80% cell death of CTCs from six patients for at least one of three platinum-compounds (carboplatin, cisplatin and oxaliplatin), whereas drug-induced CTC cytotoxicity in the remaining four patients failed to reach the 80% cut-off chemosensitivity value.

CTCs were also assayed for epidermal growth factor receptor (EGFR), vascular endothelial growth factor receptor (VEGFR), tumour protein p53 (p53), multidrug resistance (MDR1), thymidylate synthase (TYMS), dihydrofolate reductase (DHFR), serine hydroxy-methyltransferase 1 (SHMT1), DNA excision repair protein (ERCC1), and glutathione S-transferases (GST) mRNA expression (Table 3) and, based upon a CTC VEGF expression level >65% compared to peripheral blood mononuclear cells (PBMCs), bevacizumab was selected and administered to two patients, and one patient with BRCA1 mutation received rucaparib.

### 2.3. RECIST 1.1 Tumour Responses Following HAP

RECIST 1.1 tumour responses to the drugs selected by CTC chemosensitivity assay in the 10 recurrent EOC patients submitted for HAP, are presented in Table 1, which also contains the clinical characteristics of the patient cohort. One patient exhibited a complete response (CR), one patient exhibited a partial response (PR) and stable disease (SD) characterised the remaining eight patients (SD). Following multidisciplinary treatments, PFS ranged from 3 to 84 months (median 8 months, interquartile range 4–9 months) and OS times ranged from 8 to 84 months (median 15 months, interquartile ranges 12–24 months) (Table 1).

### 2.4. CTC Chemosensitivity Test Accuracy in Relation to “two-drug” HAP

Positive (complete or partial) and negative (stable disease or progression) RECIST 1.1 responses, following “two-drug” HAP, associated with CTC chemosensitivity of >80% or < 80% drug-induced cell death, are displayed in Table 4. A 100% sensitivity value was observed for complete or partial RECIST 1.1 responses, following 2 drug HAP selected by > 80% CTC chemosensitivity, and a specificity value of 50% was observed for RECIST 1.1 disease responses characterised as stable or in progression, following 2 drug HAP selected by <80% CTC chemosensitivity. A 33% positive predictive value (PPV) for a positive RECIST 1.1 response was associated with >80% CTC chemosensitivity to both drugs and a 100% negative predictive value (NPV) for a negative RECIST 1.1 response was associated with CTC chemosensitivities of <80%. The overall of value of CTC chemosensitivity tests in predicting the patient response to 2 drug HAP (accuracy value) was 60% and was calculated from the ratio of positively and negatively-corrected classified patients, using the chemosensitivity cut-off value of 80%, RECIST 1.1 criteria and the number of patients treated with 2 drug HAP.

### 2.5. Treatments Following HAP, and Patient Follow-up

Additional multidisciplinary treatments were also based upon the results from CTC chemosensitivity assays, CTC gene expression profiles and BRCA mutational status. Based on gene expression assays, two patients received bevacizumab targeted-therapy (Patients 3 and 10). Patient 3 exhibited a CR of 48-month duration, subsequently received rucaparib for 36 months, in accordance with a mutated BRCA status. This patient is still alive and continues to exhibit a CR. In contrast, patient 10 exhibited a PR of 15-month duration and unfortunately died at 24 months. Three patients with locoregional and distant relapsed disease received systemic therapy, one patient received both locoregional and systemic therapy and four patients received best supportive care, only (Table 1). The three patients with locoregional relapsed disease received additional locoregional treatments based on CTC chemosensitivity assays. Of the 10 patients enrolled in this cohort study, nine died as the result of disease progression, one remains alive today with no evidence of disease and the median OS for the cohort was 15 months (interquartile ranges 12–24 months).

## 3. Discussion

In this study, we report that the methods employed for blood sampling, storage, transport and subsequent CTC purification and enrichment are feasible and reproducible, and that the subsequent CTC chemosensitivity and gene expression assays employed to select chemotherapeutic strategies are predictive of therapeutic response. Our results confirm that the numbers of EOC CTCs purified from liquid biopsies, expanded by temporary culture, are sufficient for flow cytometry-based Annexin V-PE chemosensitivity assays, which predict RECIST 1.1 responses to “two-drugs” locoregional HAP with 60% accuracy.

The interval between blood sampling and quantitative reverse transcription polymerase chain reaction (qRT-PCR) analysis did not exceed 80 h, previously reported to minimise alterations in gene and protein expression [41] and metastatic EOC CTC numbers in 10 liquid biopsies were greater than the 5 CTCs/mL cut-off value required for chemosensitivity and tumour gene expression analysis [41]. In our study, CTC detection rate was 100%, which is significantly higher than previous reports of CTCs purified from liquid biopsies obtained from patients with primary tumour [30] and can potentially be explained by the increased numbers of CTC that would be expected to associate with the advanced stage IIIc and IV EOC metastatic tumour burden.

Considering the relatively low numbers of CTCs isolated from individual patients, isolates were expanded in appropriate media, at 37 °C and 5% CO_2_, to densities sufficient for chemosensitivity assays. This raises the possibility for phenotypic alteration and enrichment of subpopulations less representative of the bulk tumour. Since, less-differentiated cancer stem cell-like subpopulations, present in the vast majority of CTC isolates [42], expand more rapidly than more differentiated CTC counterparts [43], expanded CTC cultures are more likely to represent more aggressive components of the bulk tumour and, therefore, a more relevant cell population for predictive oncotherapy. Expanded cultures, however, did not exhibit phenotypic or genotypic alterations, assessed using short tandem repeats (STRs) or by analysing biomarker mRNA and protein expression prior to and post expansion.

Flow cytometry Annexin V-PE CTC chemosensitivity assays provided important information on several chemotherapeutic agents but in contrast to tissue-validated chemosensitivity assays, do not preserve cell-to-cell or cell-to-matrix interactions [3]. For this reason, a cut-off value of >80% cell death was chosen for drug selection based upon CTC chemosensitivity, which is far higher than the > 30% cell death cut-off normally used in predictive chemosensitivity assays employing tumour tissues that better resemble in vivo tumour architecture [3].

According to the confusion matrix displayed in Table 4, the potential of in vitro CTC chemosensitivity assays with an 80% cut-off value to predict therapeutic efficacy, evaluated by RECIST response to 2 drug HAP, resulted in values of 100% for sensitivity, 50% for specificity, 33% for PPV, 100% for PNV and a more moderate value of 60% for accuracy, all of which represent better overall values than those reported for in vitro tissue-validated chemosensitivity assays, employing a cut-off value of 30% to predict response to therapy in ovarian cancers, reported as 85.7% for sensitivity, 18.2% for specificity, 40% for PPV, 66.7% for PNV and 44.44% for accuracy [3].

With respect to tumour gene expression, CTCs from eight patients exhibited high-level p53 (>35%) expression, which is in line with reports that high P53 expression associates with minimal responses to platinum-compounds and better response to taxanes [44]. Moreover, CTCs from three patients exhibited higher ERCC1 expression (>10%) and CTCs from nine patients exhibited high-level GST expression (>10%), both are involved in platinum resistance [36,45]. Finally, CTCs from one patient exhibited high-level (≥5%) TYMS and DHFR expression, implicated in 5-fluorouracil resistance [46,47] and CTCs from eight patients showed high-level (≥50%) multi-drug resistance (MDR1) gene expression.

The scope of this EOC small cohort pilot study was to provide a feasibility evaluation of our predictive oncotherapy CTC-based approach and was not designed to evaluate treatment safety, efficacy or effectiveness [48]. This not only justifies the small sample size but also the absence of inferential statistical analysis. Clinical covariates were not considered in the analyses. Furthermore, the stringent recruitment parameters employed, including sample homogeneity, also mitigate the small sample size and are a pre-requisite for feasibility [48]. However, we caution that the percentage values for sensitivity, specificity, PPV, PNV and accuracy are indicative, at best, and should not be extrapolated to inclusion and exclusion criteria not used in this study. The efficacy data presented here are also uncontrolled and, therefore, observational and future studies, with adequate sample sizes and controlled, should aim to validate a standardised methodology for CTCs isolation, purification, enrichment, characterisation for use in chemosensitivity and tumour gene expression assays in order to elaborate a multidisciplinary treatment strategy.

## 4. Patients and Methods

### 4.1. Patients

This project was performed in accordance with the Declaration of Helsinki and approved by the ethics committee of ASL n.1, Abruzzo, Italy (10/CE/2018, 19 July, n.1419). Ten patients were prospectively enrolled from 2011 to 2016 and written informed consent was obtained from each patient. Clinical characteristics of the 10 patients with advanced stage EOC are presented in Table 1. Median age was 59 years, with an interquartile range of 56–68. All 10 patients presented with histologically diagnosed high-grade serous carcinoma and one patient exhibited a BRCA1 nonsense mutation. One patient was classified as platinum-resistant after 1st line systemic chemotherapy and was in progression after the 2nd line chemotherapy. Six patients were classified as partially platinum-sensitive and all were in progression after at least two lines of systemic chemotherapy. Three patients were classified as fully platinum-sensitive, two of whom were in progression after several lines and one after one line of systemic chemotherapy.

### 4.2. Liquid Biopsy, CTC Chemosensitivity and Tumour Gene Expression Assays

Liquid biopsies consisting of 20 mL of blood, containing 7 mL of 0.02 M ethylenediaminetetraacetic acid (EDTA), were collected from each patient in sterile 50 mL Falcon tubes, placed in impact-resistant transportation containers at 2–8 °C, transported under refrigeration and subsequently analysed [41]. CTCs were purified by layering blood samples over 4 mL of polysucrose solution (Biocoll separating solution 1077, Biochrom, Berlin, Germany), followed by centrifugation for 20 min at 2500× *g*. CTCs, peripheral blood monocytes, lymphocytes, platelets and granulocytes were collected and washed with phosphate-buffered saline (PBS, P3813; Sigma-Aldrich, Schnelldorf, Germany), incubated for 10 min in lysis buffer 154 mM NH_4_Cl (31107; Sigma-Aldrich,), 10 mM KHCO_3_ (4854; Merck, Darmstadt, Germany) and 0.1 mM EDTA in deionised water, to lyse erythrocytes, centrifuged, re-washed in PBS then sequentially incubated with magnetic beads conjugated with an antibody against leukocyte common antigen CD45 (39-CD45-250; Gentaur, Kampenhout, Belgium), followed by magnetic beads conjugated with a pan-cytokeratin antibody (pan-CK) (MA1081-M; Gentaur,) for 30 min at 4 °C. CD45 positive peripheral blood leukocytes were collected first, followed by cytokeratin positive/CD45-negative CTCs in a magnetic field, washed in PBS and cultured in 12-well plates (4430400N; Orange Scientific, Braine-l’Alleud, Belgium) in RPMI-1640 plus 10% FBS for chemosensitivity, viability and qRT-PCR assays, and compared with CD45 positive peripheral blood leukocytes (PBMCs), purified from each patient, as non-cancer cell controls. CTCs were identified by qRT-PCR, using specific primers for CK19, and other cell types excluded by RT-PCR using primers for CD31 and N-cadherin. For chemosensitivity and gene expression assays, samples contained ≥5 viable CTCs/mL.

For chemosensitivity assays, CTCs cultured in 12-well plates (3513, Corning) were treated with either: 1 μM cisplatin (P4394, Sigma-Aldrich,), 10 μM 5-fluorouracil (F6627, Sigma-Aldrich,), 1.12 μM oxaliplatin (O9512, Sigma-Aldrich,), 1 μM carboplatin (41575-94-4, Sigma-Aldrich), 5 μM irinotecan (I1406, Sigma-Aldrich), 10 nM paclitaxel (T7402, Sigma-Aldrich), 10 nM docetaxel (01885, Fluka, Munich, Germany), 5 μΜ etoposide (E1383, Sigma-Aldrich), 50 nM vinorelbine (V2264, Sigma-Aldrich), 0.5 μM topotecan (T2705, Sigma-Aldrich), 50 nM gemcitabine (G6423, Sigma-Aldrich), 1 μM doxorubicin (D1515, Sigma-Aldrich), 0.5–1 μΜ liposomal doxorubicin (300112, Sigma-Aldrich), 1 μΜ epirubicin (E9406, Sigma-Aldrich) or 2 μM mitomycin C (M4287, Sigma-Aldrich), and cell viability assessed by Annexin V-PE (559763; BD Bioscience, San José, CA, USA) flow cytometry (BD Instruments Inc., San José, CA, USA) at 24-h intervals for 6 days and the percentage of living, dead and dying cells evaluated, using BD CellQuest Software (BD Instruments Inc) and corroborated by methyl-tetrazolium (MTT), crystal violet (CVE) and Sulfo-Rodhamine B (SRB) assays [49]. The percentage of non-viable CTCs in chemosensitivity assays was calculated under non-drug and drug-treated conditions and classified as: (1) non-sensitive <35%, (2) partially sensitive 35–80% and (3) highly sensitive >80%.

For tumour gene expression assays, RNAs were purified from EOC CTCs, using RNeasy Mini Kits, as directed (74105, Qiagen, Hilden, Germany). RNAs (1 µg) were reverse transcribed using a PrimeScript RT Reagent Kit, as directed (RR037A, Takara, Beijing, China) and subjected to KAPA SYBR Fast Master Mix (2×) Universal (KK4618, KAPA Biosystems, Wilmington, MA, USA) real-time qPCR, in a final volume of 20 μL, using appropriate housekeeping genes and specific primers for epidermal growth factor receptor (EGFR), vascular endothelial growth factor receptor (VEGFR), tumour protein p53 (p53), multidrug resistance gene-ABCB1 gene (MDR1), thymidylate synthase (TYMS), dihydrofolate reductase (DHFR), serine hydroxy-methyltransferase 1 (SHMT1), DNA excision repair protein (ERCC1) and glutathione S-transferases (GST), designed using Beacon Designer 8 [50]. Following denaturation at 95 °C for 2 min, reactions were subjected to 40 PCR cycles consisting of 10 s denaturation at 95 °C and 30 s annealing at 59 °C. Melting-curve analysis was performed from 70–90 °C, with 5 s increments of 0.5 °C, at each step. All reactions were performed in triplicate, compared to template-free negative controls and evaluated by Livak relative quantification [51]. CTC gene expression was compared pre and post treatment and quantified using the following equations:
ΔCt _(threshold Cycle)_ = Ct_target_ − Ct_18SrRNA_(1)
ΔΔCt = ΔCt_(treated CTCs)_ − ΔCt_(non-cancer cells)_(2)
Relative expression level = 2^−ΔΔCt^(3)
% Gene expression = 100 × (2^−ΔΔCt^ − 1)(4)
and classified as: low over-expression (<50%) or high over-expression (>50%).

### 4.3. HAP

HAP procedures were performed under general anaesthesia, as previously described [14,52,53]. Briefly, after systemic heparinisation (150 IU heparin/kg), the common femoral artery and saphenous vein (or femoral vein, or external iliac artery and vein, when necessary) were isolated and cross-clamped. Two, previously heparinised, triple-lumen 12-F gauge balloon catheters (PfM, Cologne, Germany and Dispomedica, Hamburg, Germany), were X-ray guided into both the artery and vein and aortic and inferior cava balloons positioned, by fluoroscopy, just above the celiac trunk, above the confluence of the right hepatic vein and below the right atrium, respectively. After blocking and checking their position with contrast medium, balloons were unblocked and thighs initially blocked with pneumatic cuffs. Under temporary hyperoxygenation, chemotherapeutic agents not influenced by hypoxia and pH < 7.1 [54], such as cisplatin, carboplatin, vinorelbine, etoposide, paclitaxel and docetaxel, were introduced through the aortal catheter, as a bolus, and balloon catheters immediately blocked again. It is mandatory to inject the drugs under prior hyperoxygenation directly before balloon-blocking in order to avoid ischaemic bowel complications and for drug action [53,54]. After balloon catheter inflation, isolated perfusion of the abdominal cavity was performed by way of extracorporeal peristaltic pumps for approximately 15–20 min, under hypoxic conditions to enhance doxorubicin (and other drugs such as mitomycin) cytotoxicity [54], and at a pH of < 7.1 to enhance topotecan cytotoxicity [55]. The extracorporeal circuit (Figure 2) also contained a heating element and a hemofiltration module. Temperature loss was approximated to be 1 °C per meter of tubing and the length of the tubing was 5 m. Therefore, to ensure normothermia the element outlet port was pre-heated to a temperature of approximately 42 °C, ensuring both patient well-being and drug action. At termination of perfusion, the venous balloon was deflated prior to the arterial balloon and lower extremity tourniquets sequentially released upon attainment of a stable hemodynamic profile and the extracorporeal circuit was then subjected to chemo-filtration to reduce systemic drug access in order to reduce side effects. After chemo-filtration, catheters were withdrawn and vessels repaired by 5/0 Prolene suture. Protamine (200 IU/kg) was then injected to reverse the effects of heparin.

### 4.4. Tumour Response Criteria

Tumour responses were assessed 30–45 days following HAP, using Response Evaluation Criteria in Solid Tumours version 1.1 [56], computed tomography (CT), magnetic resonance imaging (MRI) and positron-emission tomography (PET).

### 4.5. Statistical Analysis

According to Connelly [57], a pilot study should be 10% of the sample projected for the larger parent study. Therefore, considering the small sample size in this pilot study, statistical analysis is descriptive, with purified CTC numbers, progression-free survival (PFS) and overall survival (OS) times presented as medians with interquartile ranges. The relationship between CTC chemosensitivity and response to HAP are presented without confidence intervals, as sensitivity, specificity, positive predictive value (PPV), negative predictive value (NPV) and accuracy percentages, and all computations were performed using STATA statistical software.

## 5. Conclusions

We conclude that the methodology for liquid biopsy, samples storage and transport, CTC purification and transient in vitro culture, and subsequent chemosensitivity and gene expression assays are both feasible and reproducible. However, issues that remain to be addressed, include: (i) the use of miniaturised single cell technology for low CTC numbers [58], (ii) testing with increased numbers of drugs, as on average only 10 drugs have been tested in existing reports; (iii) multiple testing prior to and following selected therapies, to take neoplastic evolution into account; (iv) analysis of primary and metastatic tumour CTCs compared to the same patient’s untreated cancer and non-cancer cells, as more appropriate controls; (v) use of more relevant 3D cell cultures; (vi) better standardisation of methodology and assays with a greater number of clinical studies to fully confirm predictive potential and importance in the selection of cancer therapy.

## Figures and Tables

**Figure 1 ijms-21-04813-f001:**
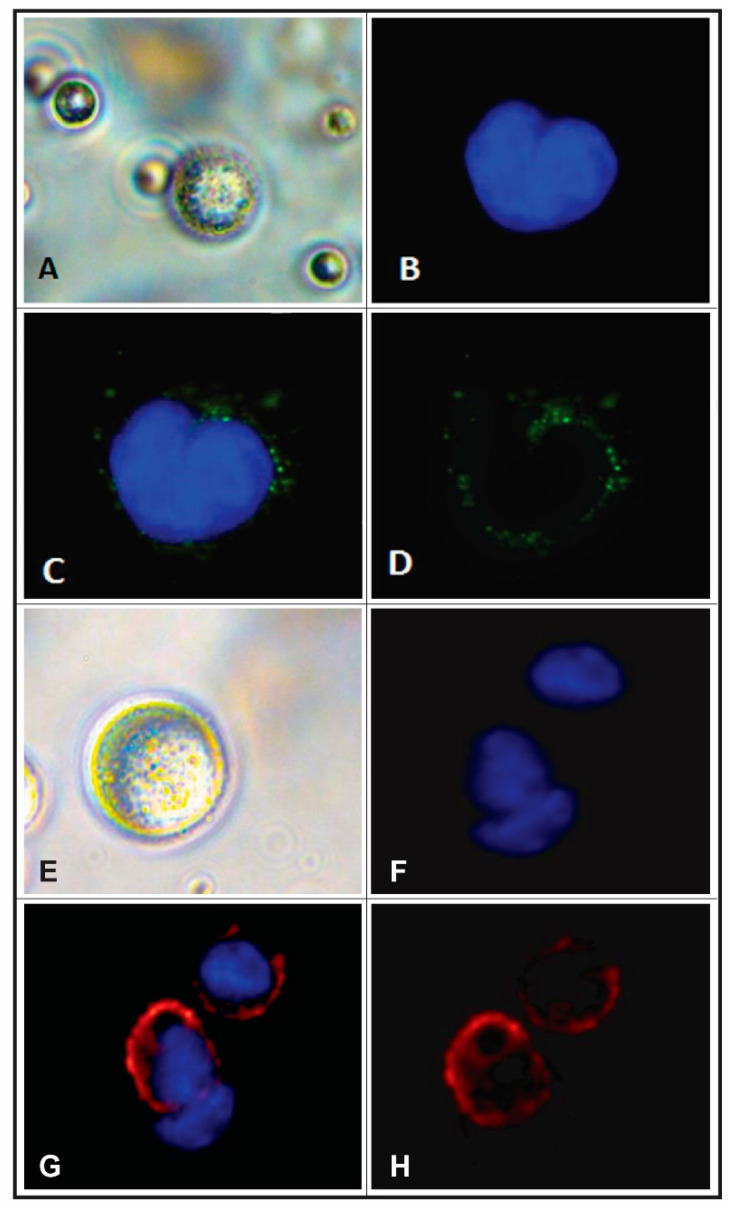
(**A**) Phase contrast micrograph demonstrating bead-isolated CTCs (40× magnitude). (**B**) Fluorescent micrograph of a DAPI (4′,6-diamidino-2-phenylindole) stained CTC nucleus. (**C**) Indirect immunofluorescence (IF) micrograph demonstrating overlapping DAPI-stained CTC nucleus (blue) and CK (cytokeratin, green) fluorescein isothiocyanate (FITC) IF, and (**D**) CTC CK IF (green), alone. (**E**) Phase contrast micrograph demonstrating bead-isolated peripheral blood leukocytes (PBL) (40× magnitude). (**F**) Fluorescent micrograph of a DAPI (4′,6-diamidino-2-phenylindole) stained PBL nucleus. (**G**) Indirect IF micrograph demonstrating overlapping DAPI-stained PBL nucleus (blue) and protein tyrosine phosphatase receptor C (CD45) (red) FITC IF, and (**H**) PBL CD45 IF (red), alone.

**Figure 2 ijms-21-04813-f002:**
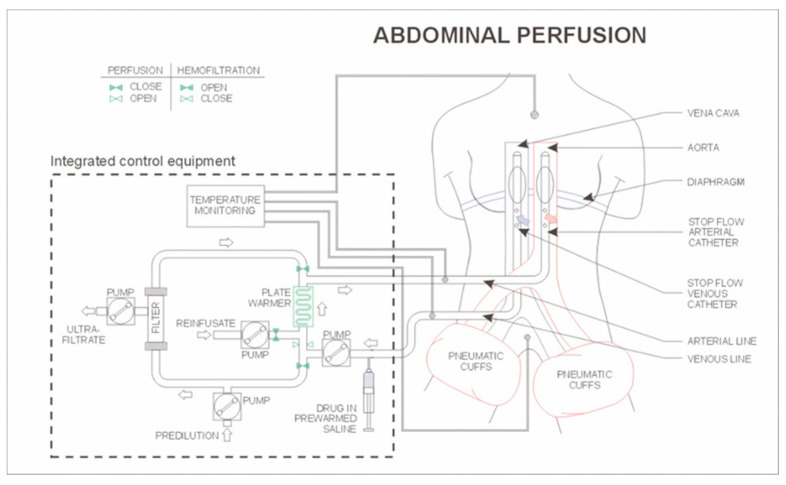
Schematic representation of hypoxic isolated abdominal perfusion (HAP) with chemo-filtration.

**Table 1 ijms-21-04813-t001:** Clinical characteristics, RECIST 1.1 tumour responses, and survival of 10 advanced epithelial ovarian cancer (EOC) patients submitted for multidisciplinary treatments.

Patient- Age	*BRCA1* Status	-FIGO -Type -Concomitant Diseases	Previous Surgery	Previous Systemic Chemotherapy	HAP [Number of Cycles] -Drugs (Dosages)	RECIST 1.1 Response	Progression Free Survival from 1st HAP	Further Therapy [Number of Cycles]	Censor at March 2020 -OS from 1st HAP
1 -58	NM	-Stage IIIC -High grade serous carcinoma	2013: Bilateral hystero-annectectomy, partial omentectomy. 2015: HIPEC (cisplatin and doxorubicin) followed by PD (peritoneal) after 3 months.	2013: 1st line with carboplatin and docetaxel followed by PD (peritoneal) after 7 months. 2014: 2nd line with cyclophosphamide and methotrexate followed by PD (peritoneal) after 4 months.	2016 HAP [3] with carboplatin (100 mg/m^2^), vinorelbine (30 mg/m^2^)	SD	6 months	Best supportive care	Dead -12 months
2 -44	NM	-Stage IIIC -High grade serous carcinoma -HIV	2010: Bilateral hystero-annectectomy, partial omentectomy. 2012: Palliative cytoreductive surgery.	2011: 1st line with carboplatin and docetaxel followed by PD (peritoneal) after 8 months. 2013: 2nd line with carboplatin, docetaxel and bevacizumab followed by PD (peritoneal) after 5 months.	2014 HAP [1] with carboplatin (100 mg/m^2^), paclitaxel (55 mg/m^2^)	SD	8 months	2015 HAP [2] with cisplatin (65 mg/m^2^), 5 FU (700 mg/m^2^)	Dead -15 months
3 -55	MT	-Stage IIIC -High grade serous carcinoma	2004: Bilateral hystero-annectectomy, partial omentectomy. 2012: Cytoreductive surgery (pelvic unresectable residual).	2012: 1st line with carboplatin and docetaxel. PD (peritoneal) after 8 years.	2013 HAP [1] with cisplatin (65 mg/m^2^), doxorubicin (30 mg/m^2^)	CR	84 months	2013 Bevacizumab (5 mg/kg) 2017: Rucaparib (250 mg)	Alive -84 months
4 -60	NM	-Stage IIIC -High grade serous carcinoma	2010: Bilateral hystero-annectectomy, partial omentectomy.	2010: 1st line with carboplatin and docetaxel followed by PD (peritoneal) after 7 months. 2011: 2nd line with cisplatin and docetaxel followed by PD (peritoneal) after 7 months. 2012: 3rd line with liposomal-doxorubicin and trabectedin followed by PD (peritoneal) after 4 months. 2013: 4th line with topotecan and gemcitabine followed by PD (peritoneal) after 9 months.	2014 HAP [2] with cisplatin (65 mg/m^2^), docetaxel (65 mg/m^2^)	SD	4 months	2014: HIPEC with cisplatin and doxorubicin	Dead -10 months
5 -65	NM	-Stage IIIC -High grade serous carcinoma	2009: Bilateral hystero-annectectomy, partial omentectomy. 2010: Omentectomy, palliative peritonectomy.	2009: 1st line with carboplatin and docetaxel followed by PD (peritoneal) after 8 months. 08/2010: Re-treatment with carboplatin and docetaxel followed by PD (peritoneal) after 8 months.	2011 HAP [2] with cisplatin (65 mg/m^2^), doxorubicin (30 mg/m^2^)	SD	9 months	2012: Liposomal-doxorubicin and trabectedin. PD after 60 months. 2017: NIPEC with Irinotecan. PD after 2 months	Dead -66 months
6 -56	NM	-Stage IIIC -High grade serous carcinoma -Diabetes	2011: Bilateral hystero-annectectomy, partial omentectomy 2012: Omentectomy, palliative peritonectomy.	2011: 1st line with carboplatin and docetaxel followed by PD (peritoneal) after 9 months. 2013: 2nd line with cisplatin and paclitaxel followed by PD (peritoneal and hepatic) after 7 months.	2014 HAP [2] with cisplatin (65 mg/m^2^), doxorubicin (30 mg/m^2^)	SD	3 months	2014: Liposomal-doxorubicin and trabectedin. PD after 6 months	Dead -12 months
7 -58	NM	-Stage IV -High grade serous carcinoma -Partial bowel obstruction -Gallbladder stones	2005: Bilateral hystero-annectectomy, partial omentectomy, aortic lymphadenectomy. 2012: Omentectomy, palliative peritonectomy. 2016: Colostomy.	2005: 1st line with carboplatin and docetaxel (allergy to carboplatin) 2005: 2nd line with cisplatin and paclitaxel followed by PD (hepatic and peritoneal) after 6 years. 2012: 3rd line re-treatment with cisplatin and paclitaxel (G2 neurotoxicity) followed by PD (hepatic and peritoneal) after 8 months. 2013: 4th line with liposomal-doxorubicin and trabectedin followed by PD (hepatic) after 10 months. 2015: 5th line with carboplatin associated to modulated electro-hyperthermia followed by PD (hepatic and peritoneal) after 9 months.	2016 HAP [1] with cisplatin (65 mg/m^2^), doxorubicin (30 mg/m^2^)	SD	8 months	Best supportive care	Dead -18 months
8 -71	NM	-Stage IIIC -High grade serous carcinoma	2013: Bilateral hystero-annectectomy, partial omentectomy, aortic lymphadenectomy. 2015: Colostomy.	2014: 1st line with carboplatin, docetaxel and bevacizumab followed by PD (peritoneal) after 9 months. 2014: 2nd line with cisplatin and paclitaxel followed by PD (peritoneal) after 7 months.	2016 HAP [1] with cisplatin (65 mg/m^2^), doxorubicin (30 mg/m^2^)	SD	9 months	Best supportive care	Dead -15 months
9 -75	NM	-Stage IIIC -High grade serous carcinoma	2013: Bilateral hystero-annectectomy, partial omentectomy, palliative peritonectomy.	2014: 1st line with carboplatin and docetaxel followed by PD (peritoneal) after 4 months. 2014: 2nd line with cisplatin and paclitaxel followed by PD (peritoneal) after 3 months.	2015 HAP [1] with etoposide (30 mg/m^2^), paclitaxel (55 mg/m^2^)	SD	4 months	Best supportive care	Dead -8 months
10 -68	NM	-Stage IIIC -High grade serous carcinoma	2008: Bilateral hystero-annectectomy, partial omentectomy. 2010: Omentectomy, palliative peritonectomy.	2010: 1st line with carboplatin and docetaxel followed by PD (peritoneal) after 7 months. 2011: 2nd line with cisplatin and paclitaxel followed by PD (peritoneal) after 9 months.	2012 HAP [1] with vinorelbine (30 mg/m^2^), topotecan (1.5 mg/m^2^)	PR	15 months	2012 Bevacizumab (5 mg/kg)	Dead -24 months

*BRCA1* = breast cancer type 1 susceptibility gene; NM = not mutated type; MT = mutated type; FIGO = International Federation of Gynecology and Obstetrics; HAP = hypoxic abdominal perfusion; NIPEC = normothermic intraperitoneal chemotherapy; HIPEC = hyper-thermic intraperitoneal chemotherapy.

**Table 2 ijms-21-04813-t002:** Liquid biopsy circulating tumour cells (CTCs) chemosensitivity assays.

Pt.	IV-CTCs	5-FU (%)	Gem (%)	L-doxo (%)	Epi (%)	Doxo (%)	MMC (%)	Eto (%)	Carbo (%)	Cis (%)	Ox (%)	Paclit (%)	Doce (%)	Vino (%)	Topo (%)	Iri (%)
1	9.6/mL, SD +/- 0.3 cells	24	76	68	56	58	45	84	83	55	35	58	63	95	70	44
2	16.8/mL, SD +/- 0.3 cells	81	21	20	28	26	40	25	81	70	50	80	65	32	61	43
3	6.9/mL, SD +/- 0.3 cells	77	50	81	77	80	52	71	70	81	52	55	52	67	60	55
4	9.4/mL, SD +/- 0.3 cells	75	82	60	65	65	60	70	75	82	65	70	75	60	75	65
5	9.4/mL, SD +/- 0.3 cells	91	22	86	42	50	35	23	52	82	61	38	42	64	62	82
6	9.8/mL, SD +/- 0.3 cells	92	25	88	42	50	36	24	53	67	61	38	45	64	62	82
7	9.6/mL, SD +/- 0.3 cells	40	70	82	65	80	22	70	65	80	60	70	65	55	40	40
8	8.2/mL, SD +/- 0.3 cells	25	90	64	43	91	53	44	64	60	38	75	82	44	82	60
9	8.4/mL, SD +/- 0.3 cells	38	26	44	23	35	47	91	22	24	21	92	58	48	36	38
10	8.2/mL, SD +/- 0.3 cells	31	24	41	42	53	46	62	58	52	28	62	58	88	91	62

Pt. = patient; IV-CTCs = isolated viable circulating tumour cells; 5-FU = 5 fluorouracil; Gem = gemcitabine; L-doxo = liposomal doxorubicin; Epi = epirubicin; Doxo = doxorubicin; MMC = mitomycin; Eto = etoposide; Carbo = carboplatin; Cis = cisplatin; OX = oxaliplatin; Paclit = paclitaxel; Doce = docetaxel; Vino = vinorelbine; Topo = topotecan; Iri = irinotecan; SD = standard deviation.

**Table 3 ijms-21-04813-t003:** Liquid biopsy CTC tumour gene expression assays.

Pt.	EGFR (%)	VEGFR (%)	p53 (%)	*MDR1* (%)	*TYMS* (%)	DHFR (%)	SHMT1 (%)	ERCC1 (%)	GST (%)
1	55	50	75	58	0	0	0	0	16
2	45	45	35	65	0	0	0	0	5
3	60	75	10	50	0	0	0	0	20
4	45	60	15	60	0	0	0	0	10
5	55	65	45	64	0	0	0	0	14
6	55	55	45	63	0	0	0	0	12
7	55	55	35	55	25	10	0	10	20
8	40	40	60	60	0	0	0	0	10
9	40	55	55	70	0	0	0	25	10
10	55	65	65	46	0	0	0	26	10

Pt. = patient; EGFR = epidermal growth factor receptor; VEGFR = vascular endothelial growth factor receptor; p53 = cellular tumour antigen p53; *MDR1* = multidrug resistance gene (ABCB1 gene); *TYMS* = thymidylate synthase gene; DHFR = dihydrofolate reductase; SHMT1 = serine hydroxy-methyltransferase 1; ERCC1 = DNA excision repair protein; GST = glutathione S-transferases.

**Table 4 ijms-21-04813-t004:** Positive (complete or partial) and negative (stable disease or progression) RECIST 1.1 responses to 2-drug HAP, selected by liquid biopsy CTC chemosensitivity assay and associated with either > 80% (positive) or ≤ 80% (negative) CTC death, for both drugs.

	RECIST 1.1 Response		
Chemosensitivity of CTCs	Positive (CR + PR)	Negative (SD + PD)	Total
Positive (>80%)	2	4	6
Negative (≤80%)	0	4	4
Total	2	8	10

CTCs = circulating tumour cells; CR = complete response; PR = partial response; SD = stable disease; PD = progressive disease.

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
