# Peer review of "A Pilot Study of the Predictive Potential of Chemosensitivity and Gene Expression Assays Using Circulating Tumour Cells from Patients with Recurrent Ovarian Cancer"

_ijms, 2020, doi:10.3390/ijms21134813_

Round 1

Reviewer 1 Report

The authors report results of a study that employs circulating tumor cells isolated from 20 mL of blood obtained from patients with recurrent, advanced epithelial ovarian cancer for in vitro chemosensitivity and gene expression analyses. This patient specific profiling of cancer cells is proposed as a means to guide selection of chemotherapeutic agents for subsequent hypoxic isolated abdominal perfusion therapy. The findings are considered in the context of a feasibility study demonstrating utility for the CTC chemosensitivity testing methodology described in the report.

Overall the idea is interesting and contributes to efforts to more precisely guide oncotherapy based on assessments of patient-specific tumor characteristics and chemosensitivity.

There is some concern with the method. For the ten patients, a median of 9.4 cells/mL were isolated from 20 mL liquid biopsy samples. These ~200 cells were then cultured and used to perform chemosensitivity assays for 15 different chemotherapeutic agents. It is not clear to what extent the cells were expanded, or what bearing this in vivo culture would have on changing the cancer cell phenotypes or enriching for various cell subpopulations that are more or less representative of the bulk tumor. If the expansion of the CTCs was limited, then that would suggest that the cytotoxicity assays were performed using very few cells (tens?) which would present concerns as well. Much of the effort in this area is based on culture of tissue biopsies using organoid or three-dimensional culture approaches that would seem to better capture the tumor heterogeneity and better recapitulate the in vivo conditions and milieu.

The major concern is whether the proposed approach offers predictive potential as suggested in the title. A reasonable indicator of this would seem to be the extent to which the CTC chemosensitivity predictions were correlated with tumor response in patients. This is assessed in terms of a binary “positive” or “negative” response according to tumor RECIST 1.1 metrics, indicating respective complete/partial response or stable/progressive disease. According to the Results Section 2.3 and confusion matrix given in Table 3, “100% sensitivity” and “100% negative predictive value” are observed. It is vaguely reassuring that there was perfect correlation between samples with the lowest in vitro CTC chemosensitivity and poor therapeutic response. Nevertheless, the issue remains that for the 6/10 patients that did exhibit >80% CTC chemosensitivity, only 2 exhibited positive therapeutic response as evident in the “33% positive predictive value” observed for patients treated according to the results of the CTC chemosensitivity screening.

As a feasibility study, this paper is the fourth study to employ this approach. Previous, similar studies have been performed in unresectable recurrent rectal cancer, recurrent cutaneous melanoma with locoregional pelvic metastases, and unresectable colorectal cancer with liver metastases as reported in REFS 38, 39 and 40. The reporting here in the context of ovarian cancer is novel, otherwise the CTC isolation, in vitro assay for agent selection, and infusion therapy are very similar to what has been previously reported by many of the same authors.

Author Response

IJMS-805148

Reply to Reviewer 1

Reviewer: There is some concern with the method. For the ten patients, a median of 9.4 cells/mL were isolated from 20 mL liquid biopsy samples. These ~200 cells were then cultured and used to perform chemosensitivity assays for 15 different chemotherapeutic agents. It is not clear to what extent the cells were expanded, or what bearing this in vivo culture would have on changing the cancer cell phenotypes or enriching for various cell subpopulations that are more or less representative of the bulk tumor. If the expansion of the CTCs was limited, then that would suggest that the cytotoxicity assays were performed using very few cells (tens?) which would present concerns as well.

Authors: thanks for the effort in evaluating and reviewing our study.

We added a new paragraph in Discussion based on your considerations.

3.Discussion, line 255 of the revised version: “Considering the relatively low numbers of CTCs isolated from individual patients, isolates were expanded in appropriate media, at 37oC and 5% CO2, to densities sufficient for chemosensitivity assays. This raises the possibility for phenotypic alteration and enrichment of subpopulations less representative of the bulk tumour. Since, less-differentiated cancer stem cell-like subpopulations, present in the vast majority of CTC isolates [42], expand more rapidly than more differentiated CTC counterparts [43], expanded CTC cultures are more likely to represent more aggressive components of the bulk tumour and, therefore, a more relevant cell population for predictive oncotherapy. Expanded cultures, however, did not exhibit phenotypic or genotypic alterations, assessed using Short Tandem Repeats (STRs) or by analyzing biomarker mRNA and protein expression prior to and post expansion.”

Consequently, we added 2 new references (42,43), we changed the list of references, and we changed numeration of references in the text.

3.Discussion, line 259 of the revised version: “[42]”

3.Discussion, line 260 of the revised version: “[43]”

References, line 604 to 608 of the revised version: “42.   Toloudi M, Apostolou P, Chatziioannou M, Papasotiriou I. Correlation between cancer stem cells and circulating tumour cells and their value. Case Rep Oncol 2011, 4, 44-54.

  1. Toloudi M, Ioannou E, Chatziioannou M, Apostolou P, Kiristis C, Manta S, Komiotis D, Papasotiriou I. Comparison of the growth curves of cancer cells and cancer stem cells. Curr Stem Cell Res Ther 2014, 9, 112-116.”

Discussion, line 280 of the revised version: “[44]”.

Discussion, line 282 of the revised version: “[45,]”.

Discussion, line 283 of the revised version: “[46,47]”.

Discussion, line 287 of the revised version: “[48]”.

Discussion, line 290 of the revised version: “[48]”.

4.2. Liquid biopsy, CTC chemosensitivity and tumour gene expression assays, line 341 of the revised version: “[49].

4.2. Liquid biopsy, CTC chemosensitivity and tumour gene expression assays, line 352 of the revised version: “[50].

4.2. Liquid biopsy, CTC chemosensitivity and tumour gene expression assays, line 356 of the revised version: “[51].

4.3. HAP, line 360 of the revised version: “[52,53,]”.

4.3. HAP, line 369 of the revised version: “[54]”.

4.3. HAP, line 372 of the revised version: “[53,54]”.

4.3. HAP, line 375 of the revised version: “[54]”.

4.3. HAP, line 376 of the revised version: “[55]”.

4.4. Tumour response criteria, line 390 of the revised version: “[56]”.

4.5. Statistical analysis, line 393 of the revised version: “[57]”.

  1. Conclusions, line 404 of the revised version: “[58]”.

Reviewer: The major concern is whether the proposed approach offers predictive potential as suggested in the title. A reasonable indicator of this would seem to be the extent to which the CTC chemosensitivity predictions were correlated with tumor response in patients. This is assessed in terms of a binary “positive” or “negative” response according to tumor RECIST 1.1 metrics, indicating respective complete/partial response or stable/progressive disease. According to the Results Section 2.3 and confusion matrix given in Table 3, “100% sensitivity” and “100% negative predictive value” are observed. It is vaguely reassuring that there was perfect correlation between smples with the lowest in vitro CTC chemosensitivity and poor therapeutic response. Nevertheless, the issue remains that for the 6/10 patients that did exhibit >80% CTC chemosensitivity, only 2 exhibited positive therapeutic response as evident in the “33% positive predictive value” observed for patients treated according to the results of the CTC chemosensitivity screening.

Authors: Thanks for your considerations.

We changed the 5th paragraph in Discussion as follows:

Discussion, line 271 to line 277 of the revised version: “According to the confusion matrix displayed in Table 4, the potential of in vitro CTC chemosensitivity assays with an 80% cut-off value to predict therapeutic efficacy, evaluated by RECIST response to 2 drug HAP, resulted in values of 100% for sensitivity, 50% for specificity, 33% for PPV, 100% for PNV and a more moderate value of 60% for accuracy, all of which represent better overall values than those reported for in vitro tissue-validated chemosensitivity assays, employing a cut-off value of 30% to predict response to therapy in ovarian cancers, reported as 85.7% for sensitivity, 18.2% for specificity, 40% for PPV, 66.7% for PNV and 44.44% for accuracy [3].

As concerning Reviewer’s observation: “Nevertheless, the issue remains that for the 6/10 patients that did exhibit >80% CTC chemosensitivity, only 2 exhibited positive therapeutic response as evident in the “33% positive predictive value” observed for patients treated according to the results of the CTC chemosensitivity screening.”, we precise that it is very important in the clinical setting to AVOID administration of drugs not effective but surely cytotoxic (related PNV, 100% in this study).

Reviewer 2 Report

In a pilot study, the authors conducted flow cytometry annexin V-PE CTC chemosensitivity assays to predict response to locoregional HAP chemotherapy, by using liquid biopsy (20 ml blood)-derived purified CTCs from 10 patients with stage IIIC/IV ovarian cancer. The authors have careful consideration when conducting the experiments. For example, when performing gene expression analyses in CTCs, they used PBMCs as non-cancer cell controls. However, the results in the current study are preliminary and the result presentation is mostly descriptive. The small sample size makes the findings possibly a chance finding. The predictive accuracy of 60% is moderate. Clinical covariates are not considered in analyses. Some presentations need to be re-organized. Additional comments are as follows:

  1. The title can be shortened. For example, “A pilot study of the predictive potential of chemosensitivity and gene expression assays using circulating tumor cells from patients with recurrent ovarian cancer”.
  2. It is unnecessary to provide the detailed history of tumor chemosensitivity assays (first paragraph in page 3).
  3. Please add an example image of CD45 positive peripheral blood leukocytes to Figure 1.
  4. The authors detected mRNA expression of 9 genes in CTCs. If the purpose of the detection of gene expression in this study was for prediction of therapeutic response, where were related results?
  5. Clinical characteristics of the 10 patients (Table 2) should be moved forward as the first table.
  6. Please provide the full names of the abbreviations at their first appearance, for example, IF in the Abstract.
  7. Please delete “≈” in the main text.
  8. Please rephrase the sentence in lines 180-183 in page 4.
  9. Use Table 1 and Table 2 instead of Table 1A and Table 1B. Add the full name of SD to the footnote of related table.

Author Response

IJMS-805148

Reply to Reviewer 2

- Reviewer: In a pilot study, the authors conducted flow cytometry annexin V-PE CTC chemosensitivity assays to predict response to locoregional HAP chemotherapy, by using liquid biopsy (20 ml blood)-derived purified CTCs from 10 patients with stage IIIC/IV ovarian cancer. The authors have careful consideration when conducting the experiments. For example, when performing gene expression analyses in CTCs, they used PBMCs as non-cancer cell controls. However, the results in the current study are preliminary and the result presentation is mostly descriptive. The small sample size makes the findings possibly a chance finding. The predictive accuracy of 60% is moderate. Clinical covariates are not considered in analyses. Some presentations need to be re-organized. Additional comments are as follows:

Authors: thanks for the effort in evaluating and reviewing our study.

We changed Discussion, based on your considerations, as follows:

Discussion, line 274 of the revised version: “a more moderate value of”.

Discussion, line 288 of the revised version: “Clinical covariates were not considered in the analyses.”

We made changes based on your additional comments as follows:

Title has been changed. Page 1, line 2 of the revised version: “A pilot study of the predictive potential of chemosensitivity and gene expression assays using circulating tumor cells from patients with recurrent ovarian cancer”.

- Reviewer: It is unnecessary to provide the detailed history of tumor chemosensitivity assays (first paragraph in page 3).

Authors: Introduction has been changed deleting detailed history of tumor chemosensitivity assay.

Page 3, line 99 to 107 of the revised version: “In addition, precision oncotherapy based upon tumour chemosensitivity assays has been under evaluation as an alternative therapeutic approach for the treatment of platinum-resistant EOC. Empiric therapies are chosen from the current literature based upon outcomes achieved for a particular tumour-type with single and combinations of chemotherapeutic agents, whereas drug-selection based upon chemosensitivity assays takes cues from the sensitivity of tumour tissues, tumour cell cultures [15-27] or purified circulating tumour cells (CTCs) to a panel of chemotherapeutic agents in in vitro cytotoxicity assays. In addition to these approaches, important information predicting a potential drug-response can also be gleaned from standard immunohistochemical, gene expression and transcription profiling of non-viable tumour tissues.”

- Reviewer: Please add an example image of CD45 positive peripheral blood leukocytes to Figure 1.

Authors: Figure 1 has been changed with addition of CD45 positive peripheral blood leukocytes images.

- Reviewer: The authors detected mRNA expression of 9 genes in CTCs. If the purpose of the detection of gene expression in this study was for prediction of therapeutic response, where were related results?

Authors: treatment related results based on gene expression assays have been added, as follows:

2.5. Treatments following HAP, and patient follow-up, line 227 to 232 of the revised version: “Additional multidisciplinary treatments were also based upon the results from CTC chemosensitivity assays, CTC gene expression profiles and BRCA mutational status. Based on gene expression assays, two patients received bevacizumab targeted-therapy (Patients 3 and 10). Patient 3 exhibited a CR of 48-month duration, subsequently received rucaparib for 36 months, in accordance with a mutated BRCA status. This patient is still alive and continues to exhibit a CR. In contrast, patient 10 exhibited a PR of 15-month duration and unfortunately died at 24 months.”

- Reviewer: Clinical characteristics of the 10 patients (Table 2) should be moved forward as the first table.

Authors: We changed as follows:

Results, line 150 to 152 of the revised version: “2.1. Biological and clinical characteristics of 10 advanced EOC patients

Table 1 reports biological and clinical characteristics, RECIST 1.1 tumour responses, and survival of 10 advanced stage EOC patients submitted for multidisciplinary treatments.”.

Results, line 153 of the revised version: “Table 1”.

2.3. RECIST 1.1 tumour responses following HAP, line 203 of the revised version: “Table 1”.

2.3. RECIST 1.1 tumour responses following HAP, line 208 of the revised version: “(Table 1)”.

2.5. Treatments following HAP, and patient follow-up, line 234 of the revised version: “(Table 1)”.

4.1. Patients, line 302 of the revised version: “Table 1”.

Authors: Consequently, Tables numeration has been changed as follows:

2.2. Chemosensitivity and tumour gene expression assays using EOC CTCs, line 160 of the revised version: “Table 2”.

2.Results, line 171 of the revised version: “Table 2”.

2.Results, line 193 of the revised version: “Table 3”.

2.Results, line 196 of the revised version: “Table 3”.

2.Results, line 212 of the revised version: “Table 4”.

2.Results, line 222 of the revised version: “Table 4”.

3.Discussion, line 271 of the revised version: “Table 4”.

- Reviewer: Please provide the full names of the abbreviations at their first appearance, for example, IF in the Abstract.

Authors: full names of the abbreviations at their first appearance have been provided as follows:

Abstract, line 33 of the revised version: “reverse transcription polymerase chain reaction (RT-PCR)”.

Abstract, line 34 of the revised version: “immunofluorescence (IF)”.

Introduction, line 58 of the revised version: “breast cancer type 1 susceptibility protein (BRCA)”.

Introduction, line 116 of the revised version: “reverse transcription polymerase chain reaction (RT-PCR)”.

Introduction, line 121 of the revised version: “epithelial cell adhesion molecule (EpCAM)”.

Results, line 189 of the revised version: “epidermal growth factor receptor (EGFR)”.

Results, line 190 of the revised version: “vascular endothelial growth factor receptor (VEGFR)”.

Results, line 190 of the revised version: “tumour protein p53 (p53)”.

Results, line 191 to 192 of the revised version: “multidrug resistance (MDR1), thymidylate synthase (TYMS), dihydrofolate reductase (DHFR), serine hydroxy-methyltransferase 1 (SHMT1), DNA excision repair protein (ERCC1), and glutathione S-transferases (GST)”.

Discussion, line 247 of the revised version: “quantitative reverse transcription polymerase chain reaction (qRT-PCR)”.

4.2. Liquid biopsy, CTC chemosensitivity and tumour gene expression assays, line 312 of the revised version: “ethylenediaminetetraacetic acid (EDTA)”.

Authors: Consequently, list of abbreviations has been changed.

- Reviewer: Please delete “≈” in the main text.

Authors: “≈” has been changed in the text with “approximately” as follows:

Introduction, line 47 of the revised version: “approximately”.

Introduction, line 83 of the revised version: “approximately”.

Introduction, line 90 of the revised version: “approximately”.

Introduction, line 91 of the revised version: “approximately”.

- Reviewer: Please rephrase the sentence in lines 180-183 in page 4.

Authors: we changed as follows:

2.Results, line 162 to 166 of the revised version: “Three patients with CTC chemosensitivity values of > 80% for one but not the other drug, received the agent achieving cut-off and received a second agent that did not achieving the cut-off, based on a multidisciplinary decision. The remaining patient, for whom chemosensitivity assay cut-off values failed to reach >80%, received the drug pair inducing the highest levels of cell-death.”.

- Reviewer: Use Table 1 and Table 2 instead of Table 1A and Table 1B. Add the full name of SD to the footnote of related table.

Authors: Tables numeration has been changed as follows: Table 1 (line 153 of the revised version), Table 2 (line 171 of the revised version), Table 3 (line 196 of the revised version), Table 4 (line 222 of the revised version).

The footnote of Table 2 has been changed as follows: “SD=standard deviation”.

Round 2

Reviewer 1 Report

The authors have made copious revisions to the manuscript to address the concerns of the Reviewers.  Many of my original reservations remain, and what is presented in its present form is just at the margin of acceptable. On balance, the idea is interesting and contributes somewhat to efforts to improve cancer therapy based on functional assessment of tumor characteristics.

The data are highly preliminary and provide more of a proof of concept for the workflow rather than a demonstration of utility for the approach and its intended purpose. In this regard, novelty is diminished by the publication of at least three similar papers by the same authors, albeit in the contexts of different cancer types.

Reviewer 2 Report

The manuscript has been substantially revised. The authors carefully and seriously addressed each comment and issue raised by the reviewers. They discussed the limitations of the current study as well. As a small pilot study, the presentation is appropriate and the overall manuscript is acceptable. No additional comments.